# Monthly Abundance Patterns and the Potential Role of Waterbirds as Phosphorus Sources to a Hypertrophic Baltic Lagoon

**Rasa Morkūnė [1],\*, Jolita Petkuvienė [1], Modestas Bružas [1], Julius Morkūnas [1] and Marco Bartoli [1,2]** 

[1]    Marine Research Institute, Klaipeda University, 92294 Klaipeda, Lithuania; jolita.petkuviene@apc.ku.lt (J.P.); modestasbruzas@gmail.com (M.B.); juliusmorkunas@gmail.com (J.M.); marco.bartoli@unipr.it (M.B.)

[2]    Department of Chemistry, Life science and Environmental Sustainability, Parma University, 43124 Parma, Italy

\*    Correspondence: rasa.morkune@apc.ku.lt; Tel.: +370-67136264

**Abstract:** Compared to external loads from tributaries and sediment recycling, the role of waterbirds as phosphorus (P) sources in estuaries is overlooked. We performed monthly ship-based surveys of waterbird abundance in the Lithuanian part of the Curonian Lagoon, calculated their potential P excretion, and compared its relevance to the riverine inputs. Phosphorus excretion rates for the censused species were assessed accounting for variations of body weights, daily feces production and their P content, and assigning species to different feeding and nutrient cycling guilds. During the study period (March–November 2018), 32 waterbird species were censused, varying in abundance from ~18,000–32,000 (October–November) to ~30,000–48,000 individuals (June–September). The estimated avian P loads during the whole study period varied between 3.6 and 25 tons, corresponding to an area load between 8.7 and 60.7 mg P m$^{-2}$. Waterbird release of reactive P to the system represented a variable but not negligible fraction (1%–12%) of total external P loads, peaking in June–September and coinciding with cyanobacterial blooms. This study is the first in the Baltic Sea region suggesting that waterbirds are potentially important P sources to phytoplankton in large estuaries, in particular, during low discharge periods.

**Keywords:** estuary; bird; guilds; visual counts; feces; phosphorus source; algal blooms

## 1. Introduction

Phosphorus (P) total and relative availability can regulate both primary productivity and algal community composition in aquatic environments [1–3]. Early studies on P budgets and transformations addressed mostly lakes, as this nutrient was considered a limiting element for phytoplankton growth in freshwater ecosystems [4]. More recently, research in this field has expanded to estuaries and coastal lagoons, where P may regulate (or co-regulate with nitrogen) harmful algal blooms [5,6]. Past and recent research analyzed in detail two main aspects of P biogeochemistry: the inventory of point inputs from tributaries or sewage treatment plants of dissolved and particulate P forms and the reactivity of sedimentary P pools [7–9]. The latter aspect lead to the development of sequential extraction techniques and to the analysis of the redox dependence of P mobility [10]. During low discharge periods, when P inputs from the watershed decrease, sediment release may represent a major P source for pelagic production [11,12]. Comparatively, the role of other biotic components of aquatic ecosystems as regulators of P cycling has been overlooked.

Macrobiota actively circulate nutrients in aquatic ecosystems and produce measurable effects at different scales. Macroinvertebrates enhance benthic–pelagic coupling [13], fish favor sediment

resuspension, and nutrient regeneration [14], while birds have been demonstrated to increase the trophic status of small ponds [15]. Colonies of aquatic birds, for example, may represent biogeochemical hot spots through birds' feeding and excretion activities [16,17]. Waterbirds use estuaries and lagoons as resting, breeding, and stopover sites where released feces might affect nutrient cycles, primary producers growth, and trophic networks [18,19]. Phosphorus enrichment via waterbirds' guano may also induce cascade effects leading to the selection of opportunistic algal groups, excess stimulation of their production and eutrophication, contrasting ecosystem services of transitional areas as nutrient retention or removal and water purification [15,20].

Being highly mobile organisms, waterbirds link spatially distinct ecosystems, including aquatic and terrestrial, and anthropic habitats [21]. Migratory birds may transport live or dead genetic material (e.g., attached organisms, prey species and associated microbial communities) between separated ecosystems and induce structural changes in species community composition, add alien species or impact water quality and provide a wide range of ecosystem functions and services [22,23].

Large colonies of waterbirds have been demonstrated to mobilize significant nutrient amounts and affect the trophic status of occupied ecosystems. In some areas, the effect of the bird feces was proven as exceptionally important. During breeding season, nutrients from waterbird colonies may enrich oligotrophic Arctic ecosystems [24,25] and mangroves at local scales [26]. Waterbird aggregations during non-breeding periods at this point have been studied much less than breeding birds; however, they might also be an important source of nutrients (e.g., [21,27]). For example, wintering tundra swans (*Cygnus columbianus*) contribute to approximately 30% of the standard fertilizers during the irrigation period in winter-flooded paddy fields [28]. The significance of avian P tends to differ through the year, as the seasonal dynamics of the structure of waterbird communities have been observed in aquatic ecosystems [29]. Birds in aggregations during periods of breeding, roosting or over-wintering might result in a significant contribution to P global inputs, while contribution of non-colonial breeding species usually remains understudied and/or assumed to be negligible.

The characteristics of bird feces and their direct contribution to P cycle depend on a range of available food alternatives and features of feeding habitats, while the distribution of feces within spatial scales in/between ecosystems varies according to the feeding behavior of birds. Applications of feeding guilds are common when studying ecological networks, contributions to ecological process or interaction to fishery or tourisms sectors [30,31]. According to feeding habitats (inside or outside study sites) and mobility patterns, waterbirds might be external or internal recyclers of P [23,32]. Waterbirds can be seen as nutrient vectors if they feed outside the resting or nesting environment; as such, their net imported allochthonous nutrients are considered as external sources. However, aggregations of waterbirds can be also considered as recyclers if they feed, rest, and nest in the same ecosystem and simply convert through their digestion organic, unavailable nutrients associated to their food into inorganic, readily available inorganic elements [26,33].

The direct effect of birds on nutrient mobilization can be estimated by converting bird numbers into potentially generated loads from literature data on metabolic excretion rates. Nutrient inputs by birds may alleviate nutrient limitations in oligotrophic ecosystems, but it may also produce significant stimulation of primary production and algal blooms when it is concentrated in relatively small, poorly flushed areas [34]. A recent study [35] has demonstrated the different direct effects of piscivorous and herbivorous birds on water chemistry and algal growth due to the different concentrations of inorganic nutrients in feces. The diet of birds is therefore an important factor affecting feces' elemental composition.

The indirect effects of birds on aquatic ecosystems are difficult to identify as they are complex. Herbivorous birds feeding on macrophytes might remove plants that stabilize and oxidize sediments and compete with phytoplankton for nutrients. As such, their feeding might indirectly favor phytoplankton blooms and turbidity. Piscivorous birds might remove organisms that feed on large zooplankton, increasing the zooplankton pressure on phytoplankton and water transparency. Such an indirect effect can be reinforced by the removal of resuspension and excretion by fish, which can mobilize

nutrients and stimulate algal growth [36,37]. Birds feeding on macrozoobenthos indirectly decrease the capabilities of a benthic habitat to provide nutrient retention services as a consequence of bio-irrigation, sediment oxidation, and preservation of Fe (III) and Mn (IV) pools [13].

Phosphorus concentrations in the Curonian Lagoon, along the south-eastern coast of the Baltic Sea, strongly depend on riverine inputs [38]. Cyanobacteria blooms, a recurrent phenomenon in this lagoon, do not occur in spring, when P riverine inputs peak, but rather in summer, coinciding with low discharge and minimum P loads from external inputs [6,12]. In summer, high water temperatures, up to 26 °C, and calm weather conditions may favor cyanobacterial hyperblooms due to the bottom water hypoxia and redox-dependent P release from sediments [12,39]. As the Curonian Lagoon is an important site for waterbirds during both the breeding period and migration seasons, bird feces might represent another important P source. Despite waterbirds being numerous in the lagoon, neither their monthly numbers nor their potential contribution to total P inputs have been published so far, whereas only maximum numbers were investigated for protected areas designations [40]. As in other sites, this type of research has to cope with a lack of data and methodological clarity including the evaluation of bird numbers, physiological parameters as excretion rates, feces elemental composition, and species behavioral aspects, which are necessary to estimate the potential role of waterbirds in mobilizing P as compared with other sources [41].

Waterbirds might be assigned as indicators of a eutrophication process or water quality in different ecosystems, because they can respond to altered levels of nutrient inputs [42]. The abundance of waterbirds in the breeding season as a core indicator has been used to follow temporal changes in the abundance of key waterbird species and to monitor the cumulative impact of pressures on the Baltic Sea including eutrophication [43]. In this paper, the approach was different, as we monitored the monthly abundance of all waterbird species in order to estimate their potential P load to the Curonian Lagoon and contribution in sustaining algal blooms. It was hypothesized that in some seasons, in particular during summer, a low discharge period, bird feces may represent an important source of reactive P, sustaining phytoplankton growth together with external riverine loads and internal recycling from sediments.

## 2. Materials and Methods

### 2.1. Study Area: General Features and Waterbird Numbers

The study was carried out in the Curonian Lagoon, the largest in Europe, located along the south-eastern coast of the Baltic Sea (Figure 1). The investigated area extends over 413 km$^2$, within the Lithuanian territory, representing nearly 1/4 of the total lagoon surface (1584 km$^2$) (Figure 1) [44]. The Curonian Lagoon is a shallow, mostly freshwater estuary with an average depth of 3.8 m and residence time of several weeks. Water temperature varies seasonally in the range of 0–2.8 °C in winter, 2.0–15.4 °C in spring, 19.1–26.3 °C in summer, and 3.2–15.5 °C in autumn [12,45,46]. During spring, balanced nutrient availability favors diatom blooms, while in summer, N and Si limitations stimulate blooms of cyanobacteria with chlorophyll a (Chl-*a*) peaks up to 400 µg L$^{-1}$ and *Aphanizomenon* flos-*aquae* as the dominating species [6]. The main water and nutrient input to the lagoon is the Nemunas River (17–23 km$^3$ year$^{-1}$) [9]. The river water enters the lagoon in the central–eastern part and then moves to the north, towards the single connection of the lagoon with the Baltic Sea. Such a northward current is evident in the lagoon during the spring period, while it is less marked in other periods due to low river discharge [47]. As the P concentration dynamics in the Curonian Lagoon strongly depend on riverine inputs, the highest values are generally measured in winter and early spring (December–March), whereas they start decreasing in the middle of spring (April) and are low during summer. Occasionally, increases in P concentrations have been observed during summer mostly due to the decomposition of recently settled algal blooms and to the transient establishment of bottom hypoxia [12,39,47].

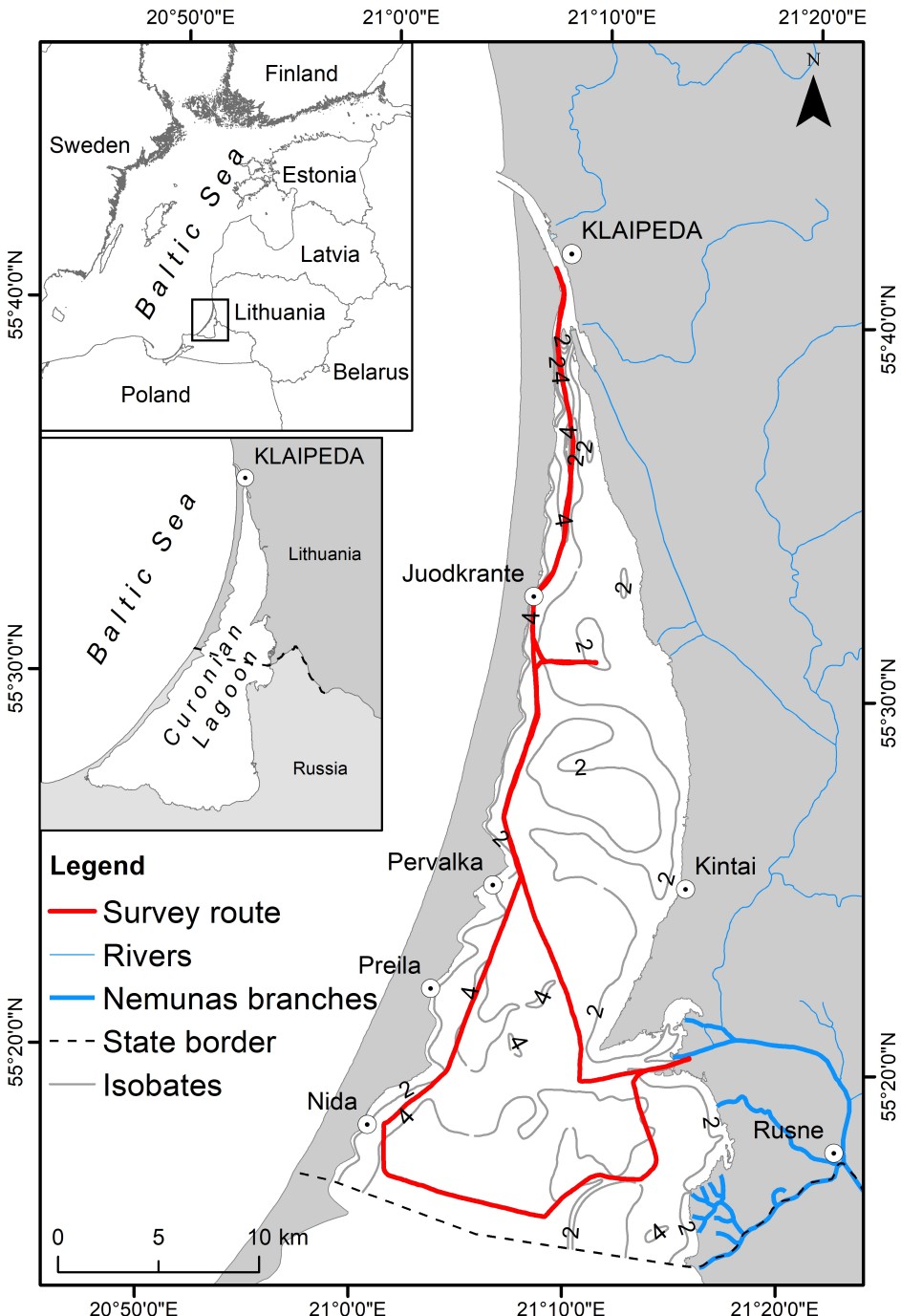

**Figure 1.** Geographical position of the study area, the Lithuanian part of the Curonian Lagoon in the South-Eastern Baltic Sea.

Bird species' composition and maximum annual abundance of the dominant species in the study area have been occasionally reported in previous monitoring activities. During spring and autumn of the last decade of XX century, the following protected bird species were registered in the lagoon: tufted duck (*Aythya fuligula*, 4000–24,500 individuals (ind.)), goosander (*Mergus merganser*, 10,000–15,000 ind.), Eurasian wigeon (*Anas penelope*, 2000–10,000 ind.), white-fronted goose (*Anser albifrons*, 6000–9000 ind.), greater scaup (*Aythya marila*, 2500–10,000 ind.), mallard (*Anas platyrhynchos*, 2000–8000), common goldeneye (*Bucephala clangula*, 2000–3500 ind.), common pochard (*Aythya farina*, 1130–4500 ind.), northern pintail (*Anas acuta*, >2000 ind.), smew (*Mergellus albellus*, 400–1000 ind.), northern shoveler (*Anas clypeata*, 100–300 ind.) (summarized data from References [40,48–52]).

At the end of summer and early autumn, concentrations of mute swans (*Cygnus olor*) reach a few thousands individuals; while Bewick's swans (*Cygnus columbianus*) and whopper swans (*Cygnus cygnus*) are less numerous (300–700 and 700–800 ind., respectively) [40,48,50,53]. Regarding breeding species in 2018, two colonies of great cormorants (*Phalacrocorax carbo*) breed at both sides of the lagoon (in the Curonian Spit and Nemunas delta; in total 5500 pairs) together with 500 pairs of grey herons *Ardea cinerea* that breed in the Curonian Spit colony (information from State Service for Protected Areas under the Ministry of Environment; [54]). A colony of black-headed gulls *Larus ridibundus* with approximately 1000 pairs breed at the island Kiaules nugara (birdringers, personal communication), in the northern part of the lagoon. Distributed breeding pairs of great crested grebes (*Podiceps cristatus*), coots (*Fulica atra*), mallards, and mute swans breed in the reed stands bordering the lagoon. Maximum abundance of waterbird species found in various reports were the basis for designation of protected status of the lagoon [40]. Bird diversity and phenology in this area was described by Castren [51]. However, monthly dynamics of waterbird abundance have not been published before this study.

*2.2. Estimation of Monthly Bird Numbers and Assignation to Guilds*

The waterbird data used in this study corresponded to the visual bird surveys performed monthly during March–November 2018 in the Lithuanian part of the Curonian Lagoon (Figure 1); surveys in the winter months (January, February, December 2018) were not performed due to the ice cover period and irregular bird occurrence in the study area. The surveys were performed by two experienced ornithologists from a vessel moving along a common waterway of the Lithuanian part of the Curonian Lagoon (as reported in References [42,55]). The monthly surveys were generally performed following the same paths in the lagoon, during the last 2 weeks of each month and each survey lasted 9–10 h during daytime. Coverage of different bird species was variable, depending on their visibility, size, color, and behavior such as hiding in reeds, performing dives, and flying to feed in surrounding ecosystems [42,56–58]. Registered data on species abundance were considered for the minimum possible number, and higher numbers as potential maximum values were assessed according to recently published information including reports and local expert knowledge.

Registered bird species were assigned to one of the four following trophic bird guilds: piscivorous, herbivorous, omnivorous or benthivorous. Considering the habitat selection, feeding, and roosting behavior, bird species were appointed to one of three following nutrient cycling guilds, according to [32] and modified in order to be applied to the study site:

A.  Net-importer guild for species which feed mainly outside the lagoon but use the lagoon as an aggregating and roosting site (e.g., geese, some gulls);
B.  Importer–exporter guild for species which feed both outside and inside the lagoon but move from/to the lagoon and surrounding territory (e.g., great cormorants);
C.  Internal recycler guild for species that spend all their time in the lagoon (e.g., waterfowl, grebes, coots, goosanders).

*2.3. Estimation of Avian Phosphorus Load*

The lowest and highest values for bird abundances and weights and P contents in feces were used to assess the ranges of P excretion rates and total loads. In the following equations, we used the lowest and highest values of each parameter to assess the minimum and maximum estimates of the P loads.

The monthly P loads for each species (Table 1) were calculated according to Equation (1):

$$M = B \times D \times R \times T \tag{1}$$

where M is the calculated monthly P loading rate (g P month$^{-1}$); B is the number of birds (ind.); D is the number of days in a particular month (days month$^{-1}$); R is the P excretion rate (g P day$^{-1}$ ind.$^{-1}$); and T is the proportion of time spent in the lagoon (%).

Daily P excretion rates for each species were calculated according to Equation (2):

$$R = F \times C/100 \tag{2}$$

where R is the calculated P excretion rate (g P ind.$^{-1}$ day$^{-1}$); F is the daily feces production (g$_{dw}$ ind.$^{-1}$ day$^{-1}$), which was calculated as percentage of body weight (3% for piscivorous and omnivorous birds, 2.25% for herbivorous birds, 3.8% for benthivorous birds [59]) or obtained from the literature if species-specific studies were available; and C is the P content in dry feces (%) as the minimum and maximum values reported in the literature.

In this study, we assumed that birds release feces randomly during the day [22]. As there were no published data at the study site, we made the following assumptions about the fraction of the day spent by birds in the lagoon:

- 100%, the entire day for little gulls, white-winged terns, black terns, little terns, common pochards, common goldeneyes, tufted ducks, greater scaups, great crested grebes, goosanders, smews, mallards, gadwalls, Eurasian wigeons, swans (mute, whooper, Bewick's), common shelducks, coots;
- 90% for common terns and Caspian terns;
- 80% for common gulls;
- 60% for black-headed gulls;
- 40% for great cormorants (data from GPS/GSM transmitters, unpublished data from local study);
- 30% for herring gulls, great black-backed gulls, greylag geese, white-fronted geese, barnacle geese, bean geese;
- 20% for grey herons, great white egrets.

### 2.4. Comparison between Avian and Riverine Phosphorus Loads

The Nemunas River represents the major water and nutrient input (>97%) to the Curonian Lagoon [6,9,12,45]. Loads of P have been monitored at a minimum monthly frequency by Klaipeda University since 2012, and sampling frequency increases in spring and autumn during high discharge events [9]. Monthly concentrations of total riverine P were also measured during 2018, and P loads were obtained by multiplying concentrations by the mean monthly river discharge. Water samples for the analyses were collected in Rusne at the Nemunas River gauging section. At the same location, discharge data for the calculations were provided by the Lithuanian Hydrometeorological Service. During 2018, other point sources of P to the Lagoon were also measured, including wastewater treatment plants from Klaipeda and Nida and small rivers, but their contribution was minor (<3%). Diffuse P sources to the lagoon are likely negligible, as there is limited agriculture along the eastern border, while the Curonian Spit is forested and most of the shoreline hosts large stands of reed belts, acting as biological filter for nutrients [9].

During 2018, at Rusne, water samples (2 L) were collected at a minimum monthly frequency for total phosphorus (TP) analyses in triplicate, integrating the whole water column by repeated Ruttner bottle sampling of surface and bottom layers. Integrated water samples were transferred to opaque HDPE bottles and transported with ice packs to a laboratory, within two hours, for subsequent analyses. Total phosphorus was determined on unfiltered water samples after digestion and oxidation of the organic P forms with alkaline peroxodisulphate acid; afterwards, concentrations of TP were quantified spectrophotometrically with the molybdate method [60].

In order to compare different sources of P on a m$^{-2}$ basis, calculated loads of monthly P excreted from waterbirds were divided by the area where bird visual counts were performed, i.e., the Lithuanian part of the Curonian Lagoon (413 km$^2$). Monthly riverine P loads were divided by the area of the lagoon (1584 km$^2$) with the assumption that water circulation distributes such loads over the whole lagoon area.

## 3. Results

*3.1. Bird Counts in the Lithuanian Part of the Curonian Lagoon*

During regular monthly counts, 32 waterbird species were observed in the period of March–November 2018 in the Lithuanian part of the Curonian Lagoon. The highest total bird number was registered in June–September (30,000–48,000 ind.), and the lowest number was registered in October–November (18,377–31,905 ind.) (Table 1). Great cormorants, grey herons, great crested grebes, goosanders, black-headed gulls, and greylag geese were breeding birds from April to July; together, they represented from 47% to 86% of all bird abundance. Coots, mute swans, and mallards formed molting aggregations that represented from 8% (July) to 25% (August–September) of total bird abundance. Aggregations of migratory birds in the lagoon dominated in March–April and August–November (Table 1).

Regarding the trophic guilds, herbivorous birds were represented by the highest number of 11 species; 9 species were piscivorous, 9 were omnivorous, and 3 were benthivorous. Among the censused birds, 20 species belonged to the internal recycler guild, 7 species were net importers, while 5 species belonged to the importer–exporter guild (Table 2).

*3.2. Phosphorus Excretion Rates by Waterbirds*

In order to estimate minimum and maximum daily P excretion rates from bird species registered during surveys, we used extremes of parameters ranges (Table 2). Firstly, the ranges of bird weights produced minimum and maximum values of daily feces production. Among the registered species, terns and smaller gulls had the smallest individual weight (down to 42 g), while the herbivorous birds were characterized by the highest weights (up to 14 kg). Minimum and maximum weights of the same species individuals varied by a factor up to 2.7. Secondly, P content in feces varied according to this pattern: herbivorous birds had the smallest P content, followed by benthivorous and omnivorous birds, whereas piscivorous birds' feces were the richest in P. The smallest and highest P content in feces of the same bird species differed by a factor of 1.7 for herbivorous birds and of 5 for omnivorous birds. When available, additional values of P content in feces were taken from the literature in order to support applied coefficients and calculated loads (Table 2).

Calculated P excretion rates (g ind.$^{-1}$ day$^{-1}$) were also provided as ranges (Table 2). As this parameter incorporated variations of bird weights, feces' amounts, and enrichment in P, its minimum and maximum values were found to be extremely variable. Regarding the feeding guilds' comparisons, herbivorous and the smallest birds of the remaining feeding guilds had the lowest daily P excretion rates, while all heaviest birds, except herbivorous, were characterized as producing the largest amounts of P on a daily basis. Considering differences in the P production rates within the same species, the minimum and maximum values varied between a factor of 1.6 and a factor of 11.3; the smallest variation was related to the majority of herbivorous birds, while the largest variation was related to benthivorous and omnivorous birds (Table 2).

**Table 1.** List of the aquatic bird species censused during monthly surveys in the Lithuanian part of the Curonian Lagoon in 2018. Minimum and maximum estimated bird numbers are reported; cell color represents the species activity.

| Breeding | Aggregations | Molting | Insignificant Numbers |
|---|---|---|---|

| Species Name | Scientific Name | Monthly Abundance, ind. | | | | | | | | |
|---|---|---|---|---|---|---|---|---|---|---|
| | | March | April | May | June | July | August | September | October | November |
| Mute swan | *Cygnus olor* | 300–350 | 682–750 | 1755–2000 | 1132–1300 | 1066–1200 | 780–1500 | 1992–2200 | 679–800 | 164–220 |
| Whooper swan | *Cygnus cygnus* | 600–900 | 900–10 | 10–20 | | | | | 60–100 | 1544–2400 |
| Bewick's swan | *Cygnus columbianus bewickii* | 100–300 | | | | | | | 20–100 | 127–500 |
| Greater white-fronted goose | *Anser albifrons albifrons* | 1300–5000 | | | | | | | | |
| Bean goose | *Anser fabalis fabalis* | 2500–4000 | | | | | | | | |
| Greylag goose | *Anser anser* | 300–500 | 150–200 | 100–200 | 70–200 | 100–200 | 150–250 | 422–700 | 398–700 | 100–300 |
| Barnacle goose | *Branta leucopsis* | 800–1400 | | | | | | | | 6–30 |
| Common shelduck | *Tadorna tadorna* | 40–50 | 10–50 | 30–50 | 50–70 | 50–70 | 10–40 | | | |
| Mallard | *Anas platyrhynchos* | 4000–4000 | 317–3000 | 2413–6000 | 1150–3000 | 2290–3000 | 2940–4000 | 3000–4000 | 4635–5000 | 11,910–20,000 |
| Gadwall | *Anas strepera* | 200–300 | 70–100 | 40–50 | 30–50 | 10–50 | 20–50 | 10–50 | 5–10 | |
| Eurasian wigeon | *Anas penelope* | 500–1000 | 300–600 | 100–200 | 350–500 | 10–400 | | | | 10–120 |
| Common pochard | *Aythya ferina* | 50–100 | 140–300 | | 50–100 | 1500–2000 | 108–200 | | 1240–2000 | 12–70 |
| Greater scaup | *Aythya marila* | | 10–40 | | | | | 50–100 | 530–1000 | 70–320 |
| Tufted duck | *Aythya fuligula* | 1000–2000 | 3161–5000 | 20–40 | 40–100 | 50–100 | 75–100 | 850–1500 | 5950–8000 | 445–800 |
| Common goldeneye | *Bucephala clangula* | 2000–3000 | 2668–5000 | 100–150 | 30–50 | 10–50 | 15–50 | 46–100 | 243–600 | 1825–2500 |
| Smew | *Mergellus albellus* | 100–300 | 10–20 | | | | | | 85–120 | 163–900 |
| Goosander | *Mergus merganser* | 1800–2200 | 200–300 | 10–20 | 9–30 | 5–30 | 20–30 | 20–30 | 50–70 | 1820–3000 |
| Great crested grebe | *Podiceps cristatus* | 300–900 | 2500–3000 | 3000–5000 | 4000–5000 | 4120–5500 | 365–4000 | 2039–3000 | 1067–3200 | 4–20 |
| Great cormorant | *Phalacrocorax carbo sinensis* | 5000–6000 | 6014–8000 | 8515–12,300 | 11,172–12,300 | 4697–12,500 | 14,073–26,300 | 22,059–30,000 | 1928–3500 | 159–200 |
| Great white egret | *Ardea alba* | | 40–80 | 35–100 | 15–70 | 18–100 | 48–200 | 160–300 | 47–200 | 10–40 |
| Grey heron | *Ardea cinerea* | 40–40 | 80–100 | 64–150 | 33–150 | 50–200 | 54–100 | 11–50 | 30–36 | 5–15 |
| Coot | *Fulica atra atra* | | 400–1200 | 1000–1500 | 1200–2000 | 2000–3100 | 3800–4000 | 4110–5000 | 510–1000 | 20–50 |
| Black-headed gull | *Larus ridibundus* | 2000–3000 | 5199–6000 | 10,355–11,000 | 10,138–20,000 | 7389–15,170 | 2504–4000 | 880–1200 | 614–1000 | 20–30 |
| Common gull | *Larus canus* | | | | | 30–150 | 30–100 | 45–70 | 210–300 | 91–200 |
| Herring gull | *Larus argentatus* | 500–1200 | 5093–6000 | 470–700 | 76–300 | 723–800 | 1541–1600 | 149–300 | 28–50 | 134–160 |
| Great black-backed gull | *Larus marinus* | 10–20 | 20–50 | 20–50 | 76–150 | 100–200 | 61–150 | 139–200 | 48–70 | 6–30 |
| Little gull | *Hydrocoloeus minutus* | | | 1000–2000 | 150–400 | 446–1000 | 50–500 | | | |
| Little tern | *Sternula albifrons* | | | 5–10 | 10–20 | 0–20 | | | | |
| Common tern | *Hydroprogne caspia* | | 25–50 | 100–150 | 90–300 | 286–400 | 37–150 | 1–10 | | |
| Caspian tern | *Sterna hirundo* | | | | 2–10 | 10–20 | | | | |
| White-winged tern | *Chlidonias leucopterus* | | | | 50–100 | 15–100 | 40–200 | | | |
| Black tern | *Chlidonias niger* | | | 257–800 | 1014–1500 | 2127–3500 | 2412–4000 | | | |

**Table 2.** Trophic and nutrient cycling guilds of waterbirds, species weight, feces production, P content, and calculated P excretion rates. In column F, only the values in bold were used for calculations, combining extreme rates of feces production from both calculated values and values from other sources; this was done in order to have the widest possible range of feces production and potential P excretion, including true rates.

| Species Common Name | Trophic Guilds * | Nutrient Cycling Guilds ** | Weight [1] (kg Individual$^{-1}$) | Feces Production (g$_{dw}$ Individual$^{-1}$ Day$^{-1}$) [3] (Values in Bold Were Used for Calculations) | | P Content in Feces (%) | | Calculated P Excretion Rate (g P Individual$^{-1}$ Day$^{-1}$) |
|---|---|---|---|---|---|---|---|---|
| | | | | Calculated Using % of Body Weight *** | Values from Other Sources | Used for Calculations | Other Values from Literature or Assumptions | |
| Mute swan | H | C | 7.60–14.30 | **171.0–321.8** | | 0.89 [9]–0.90 [10] | | 1.52–2.90 |
| Whooper swan | H | C | 7.40–14.00 | **166.5–315** | | 0.89 [9]–0.90 [10] | | 1.48–2.84 |
| Bewick's swan | H | C | 3.40–7.80 | **76.5–175.5** | | 0.89 [9]–0.90 [10] | | 0.68–1.56 |
| Greater white-fronted goose | H | A | 2.21–2.51 [2] | **49.6–56.5** | | 0.90 [11]–1.50 [4] | | 0.45–0.85 |
| Bean goose | H | A | 2.22–4.06 | **50.0–91.4** | 81.6 [4] | 0.90 [11]–1.50 [4] | | 0.45–1.37 |
| Greylag goose | H | A | 2.16–4.56 | **48.6–102.6** | 81.6 [4], 100 [5] | 0.90 [11]–1.50 [4] | | 0.44–1.54 |
| Barnacle goose | H | A | 1.21–2.23 | **27.2–50.2** | **58** [5] | 0.90 [11]–1.50 [4] | 1.00 [5], 1.40 [14] | 0.25–0.87 |
| Common shelduck | O | C | 0.93–1.45 | **27.8–43.5** | | 1.53–7.86 [7] | Applied value of gulls | 0.43–0.67 |
| Mallard | H | C | 0.72–1.58 | **16.2–35.6** | 16.6 [6], 27 [4] | 0.85 [8]–1.32 [6] | | 0.14–0.47 |
| Gadwall | H | C | 0.72–1.25 | **16.2–28.1** | | 0.85 [8]–1.32 [6] | Applied value of mallard | 0.14–0.37 |
| Eurasian wigeon | H | C | 0.55–1.07 | **12.4–24.1** | 16.7 [4] | 0.85 [8]–1.32 [6] | | 0.11–0.32 |
| Common pochard | H | C | 0.47–1.24 | **10.5–27.9** | | 0.85 [8]–1.32 [6] | | 0.09–0.37 |
| Greater scaup | B | C | 0.66–1.32 | **43.4–86.9** | | 1.46 [12]–6.79 [7] | | 0.36–3.39 |
| Tufted duck | B | C | 0.40–0.95 | **26.4–62.7** | | 1.46 [12]–6.79 [7] | Min - as zoobenthivorous [10], max **** | 0.22–2.45 |
| Common goldeneye | B | C | 0.71–1.25 | **46.7–82.2** | | 1.46 [12]–6.79 [7] | | 0.39–3.21 |
| Smew | P | C | 0.52–0.83 | **15.5–24.8** | | 6.80 [12]–14.32 [6] | | 1.05–3.54 |
| Goosander | P | C | 1.05–2.05 | **31.5–61.6** | | 6.80 [12]–14.32 [6] | As piscivorous cormorants | 2.14–8.82 |
| Great crested grebe | P | C | 0.57–0.81 | **17.0–24.4** | | 6.80 [12]–14.32 [6] | | 1.16–3.49 |
| Great cormorant | P | B | 1.67–2.69 | 50.2–80.6 | **27** [6], 45 [4] | 6.80 [12]–14.32 [6] | 7.90 [13], 12.40 [10] | 1.84–10.00 |
| Great white egret | P | B | 0.81–0.94 | **24.4–28.1** | | 6.80 [12]–11.47 [6] | | 1.66–3.22 |
| Grey heron | P | B | 1.02–2.07 | **30.6–62.2** | | 6.80 [12]–11.47 [6] | | 2.08–7.13 |
| Coot | O | C | 0.61–1.20 | 18.3–**36.0** | **13.5** [4] | 1.53–7.86 [7] | Applied value of gulls | 0.21–0.55 |
| Black-headed gull | O | A | 0.20–0.33 | **5.9–9.8** | 10.4 [7], **12.2** [8] | 1.53–7.86 [7] | | 0.09–0.96 |
| Common gull | O | A | 0.29–0.55 | **8.7–16.6** | | 1.53–7.86 [7] | | 0.13–1.30 |
| Herring gull | O | A | 0.72–1.39 | **21.5–41.6** | 24.4 [7] | 1.53–7.86 [7] | 1.62 [6], 1.87 [4], 2.23 [7] | 0.33–3.27 |
| Great black-backed gull | O | C | 1.03–2.27 | **31.0–68.2** | | 1.53–7.86 [7] | | 0.47–5.36 |
| Little gull | O | C | 0.09–0.16 | **2.6–4.9** | | 1.53–7.86 [7] | | 0.04–0.38 |
| Little tern | P | C | 0.05–0.06 | **1.5–1.9** | | 6.80 [12]–14.32 [6] | | 0.10–0.27 |
| Common tern | P | B | 0.10–0.15 | **3.1–4.4** | | 6.80 [12]–14.32 [6] | As piscivorous cormorants | 0.21–0.62 |
| Caspian tern | P | B | 0.57–0.78 | **17.2–23.5** | | 6.80 [12]–14.32 [6] | | 1.17–3.36 |
| Black tern | O | C | 0.06–0.07 | **1.8–2.2** | | 1.53–7.86 [7] | Applied value of gulls | 0.03–0.17 |
| White-winged tern | O | C | 0.04–0.07 | **1.3–2.0** | | 1.53–7.86 [7] | Applied value of gulls | 0.02–0.16 |

Letters in the header clarify Equations (1) and (2). * Trophic guilds: P—piscivorous, O—omnivorous, H—herbivorous, B—benthivorous. ** Nutrient cycling guilds: A—Net importers, B—Importer-exporters, C—internal recyclers. *** Amount of dry feces as a percentage of body weight per day [3]: 3% for piscivorous and omnivorous birds, 2.25% for herbivorous birds, 3.8% for benthivorous birds. **** Considering the difference in P concentrations between Bream *Abramis brama* (27 mg g$^{-1}$) and Chironomidae larvae (12.8) [6], the max P content in feces of piscivorous birds was divided by 2.1 and used as a maximum rate for benthivorous birds. [1] All bird mass from Dunning et al. [61]; [2] mass of greater white-fronted goose from Ely et al. [62]; [3] Sanderson and Anderson [63] and Gould and Fetcher [64] at Scherer et al. [59]; [4] Scherer et al. [59]; [5] Kear [65]; [6] Marion et al. [66]; [7] Hahn et al. [23]; [8] Gwiazda et al. [41]; [9] Somura et al. [28]; [10] Petkuvienė et al. [35]; [11] Rönicke et al. [67]; [12] Hatano et al. [68]; [13] Gwiazda [69]; [14] Purcell [70].

### 3.3. Phosphorus Load by Waterbirds in the Curonian Lagoon

The total P load excreted by waterbirds during the study period (March–November 2018) was estimated upscaling daily to monthly rates and then integrating over the study period. It varied between 3.6 and 25 tons, and it was equivalent to 8.7–60.7 mg P m$^{-2}$. Among waterbirds, great cormorants contributed for 46%–54% of total loading, great crested grebes and tufted ducks for 13–17%, goosanders for 6%–7%, mute swans for 4%–11%, black-headed gulls for 1.8%–4.3%, and mallards for 3%–3.8%. Mentioned species contributed 90%–93% of total avian P load during the study period (Figures 2 and 3). Monthly P amounts provided by all registered species are reported in Table S1.

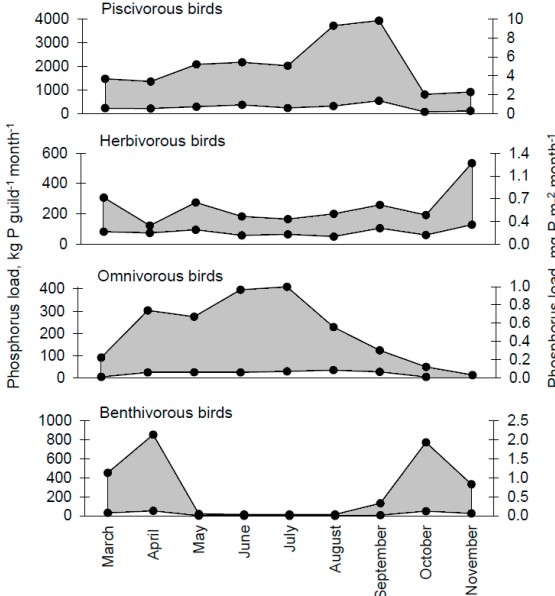

**Figure 2.** Monthly contributions of bird trophic groups to total avian P production in the Curonian Lagoon. Grey area represents ranges of possible loads. Note that the scales are different for the figures.

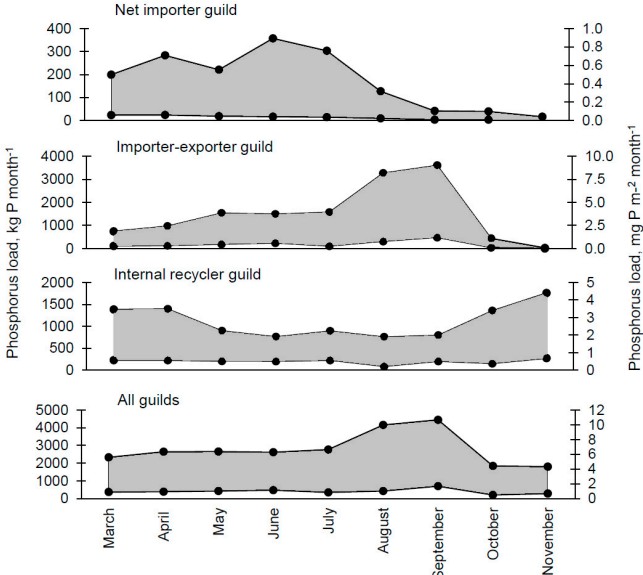

**Figure 3.** Monthly P loads produced by different nutrient cycling guilds of birds. Grey area represents ranges of possible loads. Note that the scales are different for the figures.

### 3.3.1. Trophic Guilds

Piscivorous birds contributed higher P loads to the surface of the Curonian Lagoon than other trophic guilds, in particular during summer. Their excretion accounted for 52%–68% of the total in March–April, 71%–89% in May–September and 44%–51% in October–November (Figure 2). The highest amounts of P excreted by piscivorous birds were calculated for August and September with 336–3712 and 559–3928 kg P month$^{-1}$, respectively. Monthly P areal load by piscivorous birds was estimated to vary between 0.6 and 3.6 mg P month$^{-1}$ m$^{-2}$ in March and April, 0.2–2.2 mg P month$^{-1}$ m$^{-2}$ in October and November, and 0.8–9.5 mg P month$^{-1}$ m$^{-2}$ in August and September (Figure 2).

Benthivorous birds sustained a relevant fraction of total avian P load in spring and autumn, varying between 8.5 and 32.3% in March–April and between 9.0 and 42.2% in October–November. Excretion by benthivorous birds peaked in April and October, with 52.6–852.4 and 50–771.8 kg P month$^{-1}$, respectively, while their abundance and contribution to P loads to the lagoon were low during the period of May–September (Figure 2). Benthivorous waterbirds' contribution to areal loads varied between 0.12 and 2.1 mg P month$^{-1}$ m$^{-2}$ in April and October.

Herbivorous birds were present during the entire study period and supplied total avian phosphorus load by 50.3–534.5 kg P month$^{-1}$. In comparison to other guilds, the highest share of herbivorous birds' contribution was in November: 29.7%–45.8% of total avian P load and the highest monthly load of 0.3–1.3 mg P month$^{-1}$ m$^{-2}$ (Figure 2).

The contribution of P loads by omnivorous birds was constantly low during the study period. It varied from 0.9–13.7 kg P month$^{-1}$ in November up to 14.3–409 kg P month$^{-1}$ in June–July. In the latter months, their contribution was 5.5%–15.7% of total avian P load; during the entire period it varied between 0.1 and 1.0 mg P month$^{-1}$ m$^{-2}$ (Figure 2).

### 3.3.2. Nutrient Cycling Guilds

The importer–exporter guild dominated the other guilds with P loads of 114–1569 kg P month$^{-1}$ in March–July peaking with 322–3607 kg P month$^{-1}$ in August–September. The highest contribution to the lagoon surface was estimated in September with 1.2–8.7 mg P month$^{-1}$ m$^{-2}$.

The internal recycler guild had the most stable share through the study period and contributed to 92–1390 kg P month$^{-1}$ during the period of March–October with a slight increase in the P load in November, up to 276–1756 kg P month$^{-1}$ and an areal load of 0.7–4.2 mg P month$^{-1}$ m$^{-2}$.

The contribution by net importers was higher in the first half of the year, reaching from 15 to 283 kg P month$^{-1}$, while it was much lower from August (Figure 3). The highest load to the lagoon water was in June with 0.04–0.86 mg P month$^{-1}$ m$^{-2}$.

### 3.4. Comparison between Riverine and Avian Phosphorus Loads

The P load from river discharge varied monthly. It peaked in spring (possibly up to 134 mg P month$^{-1}$ m$^{-2}$), strongly decreased in June–July (down to 47–65 mg P month$^{-1}$ m$^{-2}$), slightly increased in August–September, and dropped again in October–November (Figure 4, Table S1). Phosphorus mobilization by birds peaked in late summer and included fractions net imported to the Curonian Lagoon by adjacent environments and fractions internally recycled. Total P loads, pooling those associated to the Nemunas river discharge and those associated to birds' excretion, followed the seasonal variation pattern of riverine P inputs. However, reactive P potentially mobilized by birds might represent from approximately 1% to 12% of the total P load during the study period. Interestingly, the highest contribution of aquatic birds to P loads occurs during June–September, coinciding with relatively small riverine P inputs.

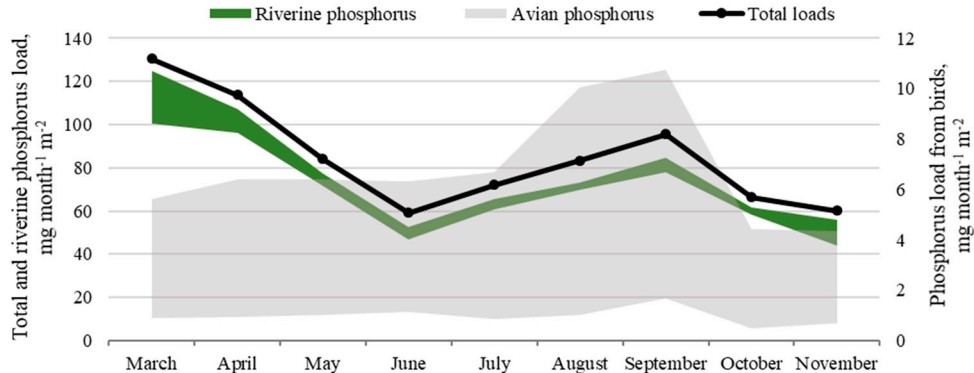

**Figure 4.** Monthly comparison of the P loads from the Nemunas River and those associated to aquatic birds excretion in the period March–November 2018 in the Curonian Lagoon.

## 4. Discussion

This work provides quantitative evidence that waterbirds may contribute significant amounts of reactive P also to large ecosystems such as estuaries. Such evidence was derived from monthly monitoring of bird numbers in the Lithuanian part of the Curonian Lagoon, the largest in Europe, and their conversion to potential P inputs through feces after detailed revision of species-specific excretion rates and P content in feces from the literature. A conservative approach was adopted, thus minimum and maximum rates and percentages were always considered in order to retrieve ranges including real loads. Calculations suggest that during the critical summer period, when temperature peaks (>23°C) and riverine P inputs decrease [12], waterbirds supply or recycle the largest fraction of their total excreted P to the aquatic ecosystem. This is mostly due to the presence of large aggregations of piscivorous great cormorants in the lagoon [54]. Such P loads, released as feces, is readily available to phytoplankton and may sustain part of the cyanobacteria summer blooms.

### 4.1. Estimations of Monthly Numbers of Waterbirds and Definition of Guilds

The Curonian Lagoon offers a diversity of habitats and attracts a high diversity and number of waterbirds. In this study, we provided monthly abundances of up to 32 species, which were assigned to different trophic and nutrient cycling guilds. The numbers of birds varied through the year and supported our hypothesis that the contribution of waterbirds to P loads undergoes wide seasonal fluctuations with a summer peak. Great cormorant was the most abundant breeding species, and it remained very abundant also during the post-breeding period till late autumn.

Previously, published bird numbers for the Curonian Lagoon were higher than the results of this work; however, the bird countings were performed by applying different methodologies from the coast surveys to counts using vessels and helicopters. Unfortunately, detailed information has not been published and only maximum numbers of birds, usually without indications of appearance periods during the year or seasonal fluctuation, have been reported (e.g., [49]). Regarding the long-term variation of waterbird abundances, nesting pairs of great cormorants have become numerous during the last decades in the close vicinity of the Curonian Lagoon [71]. Monitoring data related to other bird species staging in the lagoon during spring–autumn periods have been scarce, and we cannot infer about trends; long-term abundance dynamics of migratory birds can be described only according their population status in entire areas (e.g., Birdlife database). Concerning the effects of climate change on species abundances, the ice cover period in the semi-enclosed Curonian Lagoon has decreased by 1.6–2.3 days year$^{-1}$ over the last decades [72], a trend which is more pronounced than in other regions of the Baltic Sea. A comprehensive framework to predict the general response of bird species to climate change is still lacking [73], while mute swans, goosanders, goldeneyes, mallards, and coots were already mentioned as species affected by climate change in Lithuania [48]. As the abundance of wintering waterbirds mostly depends on the availability of ice-free water bodies [48], it is predictable

that short-distance migrants and wintering waterbirds might increase in numbers during ice-free winter periods [74]. Some data of wintering birds in the lagoon come from midwinter surveys during warmer periods in January, which might account for the estimates of goosanders (10,000 ind.), mallards (5000 ind.), and smews (500 ind.) [52]. Taken together, these sparse data suggest that less days with ice cover will lead to suitable conditions for birds to stay longer in the lagoon and obviously produce additional P loads (e.g., [70]). However, during winter, phytoplankton is limited by light and temperature.

We aggregated species into two types of bird guilds, i.e., trophic and nutrient cycling guilds, while most authors report nutrient loads only for the most abundant species [69] or use a single guild type [32,59], and they do not provide estimations for the whole bird community and seasonal comparison of P release. Our approach was to use both types of guilds in order to report avian P loads with respect to different species' diets and behavior during the study period in the Curonian Lagoon. Considering the available literature on bird feces' elemental composition, the majority of studies focused on herbivorous birds and piscivorous great cormorants, while other piscivorous birds and benthivorous species were overlooked. This might be explained by the difficulties in sampling feces from these birds considering their breeding and feeding behavior, even for birds composing large aggregations [75]. At the same time, omnivorous bird species might have significant diet variations in different ecosystems depending on the most available food sources and feeding habitats [76,77]. Regarding the high number of species in this study (i.e., 32), we made assumptions about the feces amounts and P content for unstudied birds (Table 2), similar to what other authors did in comparable studies (e.g., [23,78]).

## 4.2. Daily Phosphorus Excretion by Waterbirds

Waterbirds produce feces with high P content that, depending on bird behavior, might enrich both terrestrial and aquatic environments [23,34,79]. Even if P inputs by waterbirds have a minor importance on a global scale, they can be significant at the local scale [23]. Daily P loads for different species might differ among studies because they depend on chosen estimation methods and requested parameters. Even a simple parameter, such as bird weight, might be determined differently if authors do not consider size dimorphism and variations depending on geographical position, the stage of the annual cycle or even daytime [61,62]. In this study, we used ranges of parameters to calculate the P excretion rate for each single species registered during our surveys in the Curonian Lagoon. We obtained daily P loads for the various species by using minimum and maximum values for bird weight and P content in feces; therefore, selected coefficients combined information from body weight and diet type.

The released amount of P calculated for piscivorous cormorants ranged from 1.8 to 10 g P day$^{-1}$ ind.$^{-1}$, and published values fitted into this range with values of 2.1–3.2 [23], 3.1 [68], 3.9 [80], and 4.6 g P day$^{-1}$ ind.$^{-1}$ [66]. Grey herons and great white egrets produced 2.1–7.1 and 1.7–3.2 g P day$^{-1}$ ind.$^{-1}$, respectively, similar to the values of 3.8 g P day$^{-1}$ ind.$^{-1}$ generally reported for all herons [66], 2.3 g P day$^{-1}$ ind.$^{-1}$ reported for grey herons, and 1.5 g P day$^{-1}$ ind.$^{-1}$ reported for great white egrets [68].

The P loading rate of the abundant black-headed gulls were 0.09–0.96 g P day$^{-1}$ ind.$^{-1}$ matching with 0.23 [32,64], 0.28 [68], and 0.96 g P day$^{-1}$ ind.$^{-1}$ [41]. This load rate for herring gulls was between 0.3 and 3.27 g P day$^{-1}$ ind.$^{-1}$ matching reported values of >0.115 [30,41], 0.6 [41,81], 0.9 [42], and 1.2 g P day$^{-1}$ ind.$^{-1}$ [68]. Large great black-backed gulls might release 0.5–5.4 g P day$^{-1}$ ind.$^{-1}$ including also previously published excretion of 5.1 g P day$^{-1}$ ind.$^{-1}$ [81]. While little gull daily release of P might only be 0.04–0.38 g P day$^{-1}$ ind.$^{-1}$, in agreement with the modelled value of 0.15 g P day$^{-1}$ ind.$^{-1}$ [23]. Meanwhile, the loading rate for common gulls was 0.13–1.3 g P day$^{-1}$ ind.$^{-1}$ and incorporated the published rates of 0.3 [32,64] and 0.48 g P day$^{-1}$ ind.$^{-1}$ [68]. According to our estimations, the herbivorous mallard P load rate of 0.14–0.47 g P day$^{-1}$ ind.$^{-1}$ fitted to the reported rates of 0.18 [82] and 0.42 g P day$^{-1}$ ind.$^{-1}$ [41]. The P loading rates of other herbivorous ducks and

the majority of studied geese matched the published rates chosen according to the relative body weights [20,32,83]. Three species of benthivorous ducks might produce 0.22–3.39 g P day$^{-1}$ ind.$^{-1}$ that include species-specific values of 0.74–1.12 g P day$^{-1}$ ind.$^{-1}$ reported by Hatano et al. [68]. The P loading rates of both herbivorous and benthivorous ducks did not match with values provided by Olah [83] or Boros et al. [32], because they were also unusually low compared to other reports. As smaller and larger species with similar diets cannot have the same P excretion rates, we speculate that individual weights were not considered in References [32] and [83].

We claim that our approach to use the lowest and highest possible values of all parameters for calculations provides a possibility to integrate previously collected knowledge and local information about bird behavior and obtain reliable ranges of P loads excreted by birds to the lagoon.

### 4.3. Phosphorus Mobilization by Waterbirds and Its Relative Importance at the Curonian Lagoon Level

Considering avian P loads entering or recycled into the lagoon water, the P amounts from piscivorous birds were much higher than those from any other feeding guild for the entire period. Great cormorants contributed 34%–57% of all avian-derived P loads during breeding season in May–June, while this percentage increased up to 70%–81% in August–September (up to 7.9–8.7 mg P month$^{-1}$ m$^{-2}$) due to the added contribution of juveniles from local colonies and of migratory individuals coming from northern areas as supplementary Table S1 P sources [56].

It is generally known that despite cormorants and herons foraging in waterbodies or seas, their breeding colonies are located in surrounding habitats, which determines a continuous nutrient transfer from aquatic to terrestrial environments. During the breeding season of 2018, two cormorant colonies were present in the vicinity of the studied part of the Curonian Lagoon. Even the trees with nests were relatively distant (minimum distance was approximately 300 m) from the lagoon coast, the transfer of P from the terrestrial back to the aquatic ecosystem was possible (e.g., via runoff). Also, cormorants defecate in the lagoon water when they hunt for fish or stay on the coastal resting places. A similar, two-way aquatic–terrestrial and terrestrial–aquatic path of P loads was described for a heronry at an island in Murphy Park where the highest P concentrations in the water were determined at the peak of the breeding period [33].

The contribution of birds to nutrient cycling is demonstrated also in terrestrial ecosystems and in particular within or in the proximity of nesting areas. Allochthonous nutrient inputs can be key components that contribute greatly to explaining ecosystem productivity and dynamics [19]. The concentration of available P for plants was from 2.4 to 53.0 times higher in the soils located in areas inhabited by birds than the control areas [16]. Marion et al. [66] stated that the P concentration on areas directly occupied by colonies of herons was 42 times higher than out of the colonies. In the Curonian Lagoon, herons and cormorants do not breed over the lagoon water but spend more time in or over water during feeding and post-breeding period. However, regarding, for example, Marion et al.'s [66] findings, not only colonies but also aggregations of birds represent hotspots of nutrients. Moreover, the migration of biogenic nutrients with surface flow connects terrestrial and aquatic ecosystems and closes the cycle [16].

Considering the identified three nutrient cycling bird guilds in the Curonian Lagoon, internal recyclers represented by most herbivorous and benthivorous birds and non-colonial piscivorous birds (Table 2), dominated in the context of P release among other guilds in March–April and October–November. An importer–exporter guild of cormorants, herons, Caspian and common terns released the highest amount of P in August–September. Both importer–exporters and internal recyclers were similarly important in the context of P release from May to July. The significance of net importer guild, represented by middle-sized gulls and all geese species, was small. In similar studies, patterns of P import differed not only by the relative contribution by guilds. Regarding the characteristics of study sites, species designation to guilds might be different resulting in clear mismatches in the final estimations of P loads. The net-importer guild, represented mostly by herbivorous birds, had the most important contribution to the C, N, and P loads in intermittent alkaline soda pans in the

Carpathian Basin [32]. In our study, herbivorous birds were appointed to internal recyclers. Other species of birds, such as gulls and waterfowls, can be seen to be designated to trophic and nutrient cycling guilds similarly. Patterns of movement for geese, herons, and other birds depends on seasons, area characteristics, etc.

Our data suggest that birds' relative contribution to the lagoon's P inputs or recycling is minimum in winter and autumn due to the high discharge and P concentrations in the Nemunas River water. We have limited information on what happens to bird droppings when they reach the water: part of the reactive P can immediately dissolve into the water and part can sink and be incorporated within surface sediments [35]. Low temperatures and limited microbial activity should result in oxidized sediments where P can be retained [12]. The increase in piscivorous birds during summer can be relevant from a biogeochemical perspective, as these birds mobilize large P amounts in a period during which external inputs are low, surface sediments have limited P retention capacity, and P likely regulates cyanobacterial blooms [6,39]. Moreover, feces from piscivorous birds are richer in P than those from herbivorous birds and have been demonstrated to stimulate algal growth [35].

Our calculations indicate that P mobilized by waterbirds may represent from 1% to 12% of the load generated by the Nemunas watershed. They also suggest that such a percentage peaks during summer. Considering that a major fraction of P loads from the Nemunas river are net exported to the Baltic Sea and are not available to phytoplankton, the relative role of waterbirds as P sources could be even larger. Different studies reported a much smaller role of birds in supplying N and P to large aquatic ecosystems. However, in many areas, the role of birds was never quantified, as it was considered not relevant. Till 5–6 decades ago, when P inputs from sewage treatment plants and from agriculture were much lower, in some aquatic environments, the contribution of waterbirds as P sources was representing >90% of the annual total load (e.g., in the hypereutrophic Lake Grand-Lieu), thereafter decreasing to <10% [66]. Retrospective analyses of cyanobacteria hyperblooms in the Curonian Lagoon revealed that these events occurred also in the past and that chlorophyll a peaks up to 500 μg L$^{-1}$ with scum formation were frequent during summer [84]. As cyanobacteria fix dissolved $N_2$ and as they do not need silicon, the role of P seems central in regulating their growth [6]. Vybernaite-Lubiene et al. [9] analyzed recent trends of riverine P inputs to the Curonian Lagoon and reported that loads from the Nemunas River decreased in the last 30 years by 60%, mostly due to the improvements in water treatment and new formulation of detergents. Such a large decrease was uncoupled to a significant improvement of the lagoon water quality in terms of algal concentrations. These results support the idea that further reduction of external load may not determine a significant reduction of blooms. This may be due to the fact of three main reasons: (a) most external P loads are exported and not utilized by phytoplankton anyway; (b) waterbirds' P inputs are readily available and sustain high phytoplankton biomass, especially in summer; and (c) internal P sources as sediments supply large amounts to the water column. The latter was investigated seasonally via intact sediment incubations; results revealed that the establishment of hypoxic or anoxic conditions may favor large redox-dependent P mobilization, in particular in muddy areas of the Curonian Lagoon [12,39]. Ultimately, the importance of sediments as a P source may depend on climate change and on the number of windless days, water stratification, and high temperatures, favoring benthic oxygen shortage. Under normoxic conditions, sediments generally retain P; thus, in the absence of high discharge, the inputs from birds through feces are potentially important for phytoplankton.

Even if in this work we assumed that waterbirds spread their feces all over the lagoon surface, it is likely that much higher inputs occur in the proximity of colonies or resting places that are, generally, sheltered and shallow and, as a consequence, favorable sites for algal blooms. Experimental activities revealed that in such areas concentrations and growth rates of algal cells are higher and that sediments are chemically reduced and release large P amounts to the water column [35]. Colonies and aggregation sites are surely hot-spots for P and likely priming sites for cyanobacterial blooms due to the unbalanced nutrient stoichiometry [6,35]. This aspect is also understudied and deserves future investments with appropriate techniques as satellite remote sensing. To our knowledge, the

mechanisms of cyanobacteria hyperblooms formations are not described yet, and it is not clear whether they occur simultaneously over the whole lagoon surface or if they originate in some specific areas and rapidly spread. An interesting hypothesis is that hyperblooms create a domino effect due to the local algal growth, organic enrichment, and local anoxia and redox-dependent P release, producing an increase and self-sustainment of blooms [6,39]. We calculated that during summer, birds display the highest relative contribution to P inputs at the whole scale of the Lithuanian part of the lagoon, assuming homogeneous fertilization. However, even the effects of waterbird feces are likely patchy and close to colonies and aggregations, and these inputs are surely dominant and significantly affect water and sediment quality, algal production, and species composition.

## 5. Conclusions

This is the most detailed study providing monthly numbers of birds in the Curonian Lagoon. Our bird abundance estimations can be used as solid reference in further research in different fields, including climate change issues, conservation of biodiversity and habitats, ecosystem functioning, and biogeochemistry. The presence of piscivorous birds, especially cormorants but also goosander, grebes, and herons, in a waterbody raises discussions among the public, scientists, and environmental specialists about their effect on fish communities and the financial loss from a decrease in commercial catches (e.g., [85]). Estimations of P loads potentially released by birds reported in this and other studies may also produce public concern about the eutrophication effects of waterbirds on aquatic ecosystems. However, we are still far from understanding the complex mechanisms that regulate ecosystem functioning, including variations in fish stocks and algal blooms. More detailed analysis of the fate of feces, by means of in situ and laboratory measurements, by the implementation of ecological models, and by combination of spatial birds' distribution and satellite remote sensing of chlorophyll a are needed. The latter may reveal with appropriate scale analysis whether the presence of bird aggregations determine via nutrient mobilization hotspots of pelagic primary production. Hyperblooms of cyanobacteria represent a major threat for the studied and other Baltic lagoons; therefore, nutrient management plans require detailed knowledge not only of the inventory of nutrient sources but also of the mechanisms by which these nutrients sustain algal growth.

**Supplementary Materials:** The following are available online at http://www.mdpi.com/2073-4441/12/5/1392/s1, Table S1. Monthly phosphorus loads from river discharge and bird feces in 2018 in the Lithuanian part of the Curonian Lagoon. All numbers are provided in mg P month$^{-1}$ m$^{-2}$.

**Author Contributions:** Conceptualization and methodology, M.B. (Marco Bartoli), R.M., and J.P.; Data curation, R.M., J.P., M.B. (Modestas Bružas) and J.M.; Formal analysis and visualization, R.M. and J.P.; Writing, R.M., M.B. (Marco Bartoli) and J.P. All authors have read and agreed to the published version of the manuscript.

**Funding:** The study and manuscript preparation was made by the project "PatCHY" (No. S-MIP-17-11) supported by the Research Council of Lithuania.

**Acknowledgments:** We kindly acknowledge Irma Vybernaite-Lubiene and Mindaugas Zilius for assistance in the field and in the laboratory analysis. We gratefully thank the Lithuanian Hydrometeorological Service under the Ministry of Environment for providing Nemunas River discharge data.

**Conflicts of Interest:** The authors declare no conflict of interest.

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
