# Peer review of "Monthly Abundance Patterns and the Potential Role of Waterbirds as Phosphorus Sources to a Hypertrophic Baltic Lagoon"

_water, doi:10.3390/w12051392_

Round 1

Reviewer 1 Report

The manuscript is interesting and generally well written. This research describes impact of waterbirds on P input to large and shallow Baltic lagoon. Estimation of P input by waterbirds has been studied in small shallow waterbodies but only few studies concern large water ecosystems. So results of this study can expand existing a knowledge. The weakness of the estimation is that birds was counted only in one year. The fraction of the day spent by birds in the lagoon was not studied or estimated based on scientific data (except of cormorants).

Abstract and introduction is clear and well written. Some information should be added to material and methods. Generally results including figures are presented properly. Discussion is wide but in my opinion is too long and information is given inconsistently. This part should be ordered and rearranged.

I suggest to give more simply subtitles and order of providing information in discussion. For example:

  1. Numbers of waterbirds in seasons and P load of particular bird species.
  2. P load of waterbid assemblages and guilds into ecosystem.
  3. Importance of waterbirds in the total P load into ecosystem.
  4. Expected of waterbird abundance and algal blooms in the Curonian lagoon in the future.

Detail comments:

L. 15, 95, 100, 109, 122, 159, 238, 272, 289, 298, 321, 345, 353, 372, 386, 395, 407, 448, 449, 470, 484, 513, 520, 522, 541, 544, 589, 868,

Is "lagoon", should be "Lagoon".

L. 59

Is "Tundra swan", should be "tundra swan". Check other names.

L. 96-98

It can be impact of water temperature as well.

L. 122-127

Water temperature is a very important variable for algae development. Water temperature in the studied months in the lagoon (for example on the depth 1 m) should be added to the study area.

L. 138

Is "Northern shoveler", should be "northern shoveler".

L. 158-159

Time of survey and number of observers should be given.

L. 193

P is the daily feces production (g ind-1 day-1)

Weight of feces was given as g of dry mass. It should be written.

L. 193

Values were taken only from one paper (Scherer et al. 1995). In this paper values of daily feces were given  only for some species. Percentage of feces was calculated as a body mass of different bird groups given in wet mass and feces as dry mass. It is not proper. Moreover you should explain why such % values were taken. They do not correspond with data of Scherer et al. (1995).

Table 1.

Why Larus fuscus was not included? Explain.

Table 2

Feces production, (g ind-1 day-1)

You estimated wet mass of feces and dry mass of feces was given in the cited references.  You should add clear information.

L. 250 and L. 289

Subtitles 3.2 and 3.3 are almost the same. The title should refer to the content.

L. 322

"Guilds" is given twice. Delete.

L. 347

I suggest to change the subtitle.

L. 347-399

The all part should be moved and pasted before Expected variation of waterbird abundance in the context of climate change and mechanisms leading to algal blooms the Curonian lagoon.

L. 357

Temperature should be given.

L. 359-360

Give citation.

L. 365

Add "especially deep".

L. 400

I suggest to change the subtitle.

L. 401

Daily P loads depends mostly of diet.

L. 435

This subtitle should be changed.

L. 435-448 and L. 449-511

Parts of these two subtitles should be joined in one section.

L. 441-448

This part should be moved to material and methods.

L. 452

Is "Great cormorant", should be "great cormorant".

L. 449

Subtitle is written by bold. This subtitle should be changed.

L. 512-587

Partly it is not discussion of the results but opinion about changes in the future. Please consider to reducing of this section. 

L. 512

This subtitle should be changed. I suggest "Expected of waterbird abundance and algal blooms in the Curonian lagoon in the future".

L. 515

(Fernandez et al., 2005) should be given as a number.

L. 521-540

This part should be moved to the beginning of the discussion.

L. 592-595

This was not studied and it is not conclusion of the study. I suggest to remove these sentences.

Reviewer 2 Report

The authors investigated the P contributions of a resident and migrant waterbird community within the Curonian coastal lagoon, situated along the Baltic Sea. Recurrent cyanobacterial blooms develop in summer months of the lagoon, potentially stimulated, and likely sustained, by internal nutrient recycling from the sediments, as well as other mechanisms. Such sediment P dynamics have been more thoroughly investigated, relative to biotic factors (e.g., waterbirds), but as the authors describe, waterbirds have been shown to influence nutrient dynamics in small-scale aquatic systems. Therefore, a more thorough understanding of the waterbird community’s ecological role with respect to nutrient cycling is important for managing eutrophication.

The manuscript establishes a foundational framework for addressing waterbirds’ nutrient contributions to the Curonian lagoon, and provide an initial set of values for further assessment. As the authors acknowledge, there is substantial complexity in estimating waterbird P loads, but the manuscript makes a thoughtful effort in addressing assumptions and limitations in the available datasets. The manuscript addresses a major limitation, providing estimates of waterbird community composition and magnitude, and does a good job in guiding the reader through the calculations and assumptions required to estimate the range of guild-specific avian P loads. But with these estimates, the authors do not provide as thorough of a derivation for the riverine P load, ultimately skewing their relative comparisons. Acknowledging limitations in available nutrient data, additional efforts are required to better constrain riverine P as best as possible (item 4), and more discussion is warranted on the relative magnitude of avian P loads to internal P recycling of the system.

  1. Figure 1 – It would be helpful for the Nemunas River to be marked. Consider as well, illustrations of sampling transects/areas if it does not complicate the map.
  2. Line 177. It’s not completely clear how bird weight was determined. It would be useful for the authors to generally explain how the range of bird weights were derived from the cited studies. The range of weights are relatively variable, so I am curious as to whether or not different considerations were taken. The authors demonstrate on lines 410 – 425 that their method of calculation aligns with previous, independent estimates, so this is request is for further clarification/awareness.
  3. Line 198 – 200. How did the authors arrive at the estimate of fraction of day spent in the lagoon? Great cormorant % rely on existing literature, but Line 475-483 suggest that nesting areas are the dominant locale of excretion.
  4. Line 214 – 222. Please explain the confidence in applying 1-2 sampling points across an entire mean rate of discharge for each month. Does sampling near the mouth of the Nemunas represent nutrient concentrations delivered throughout the watershed during high discharge rates? A figure or table of intra-monthly discharge rates would be valuable to future investigators that wish to understand the sampling periodicity necessary to be constrain nutrient inputs. Additionally, long-term monthly nutrient measurements (min, max, measure of variation) would be valuable as well, if available (Vybernatite-Lubiene et al. citation on Line 375?). The authors end up with a surprisingly well-constrained riverine P load monthly and annually (Supplemental Material).
  5. Line 217 – 219. Is there information on land-use within the relevant watershed? It would be valuable to briefly explain the selected reference and why diffuse/point sources are not important (except for nesting sites?).
  6. Table 1. A comment on general organization would be useful – it appears organized by species in some manner.
  7. Table 2. ‘W’ isn’t used…is it needed? Considered moving Guild designations to Table caption.
  8. Figure 4. This figure could be misleading. Only externally derived (i.e., imported) avian P should be considered here if comparing with riverine loads. If avian P that has been internally recycled is included here, shouldn’t P resuspended from the sediments (or any other internally recycled P)?
  9. Line 364 – 368. These statements should be expanded. How much of the riverine P is considered to be exported? The term mobilized is used for avian P…is there a sense of what portion is readily available for uptake? In studies that do quantify avian nutrient supply, is there something ecologically different or is it a difference in estimation technique?
  10. Line 504 – 506. How should one consider the mobilization of P from waterbirds that feed on another biotic internal recycler (fish) of the same system? What component of avian feces should be considered ‘new’ or ‘released’ to the system, compared to fish that were also consistently releasing P?
  11. Line 582 – 584. It is not clear what this statement is made relative to. Just relative to riverine P?
  12. Line 592 – 595. Unsure the purpose of these statements? What does it have to do in relation to avian nutrient contributions?
  13. The overall discussion meandered at times and is too long – the authors should review discussion content and determine what is needed to carry the major conclusions. Some language editing is also required - it did not take away from comprehension, just slightly inhibiting the reader’s flow.
